# Asian American University Students’ Adjustment, Coping, and Stress during the COVID-19 Pandemic

**DOI:** 10.3390/ijerph20054162

**Published:** 2023-02-25

**Authors:** Jacqueline Hwang, Yi Ding, Eric Chen, Cixin Wang, Ying Wu

**Affiliations:** 1Graduate School of Education, Fordham University, New York, NY 10023, USA; 2College of Education, University of Maryland, College Park, MD 20742, USA

**Keywords:** COVID-19, pandemic, university students, Asian American, adjustment, coping

## Abstract

The COVID-19 outbreak caused global disruptions in all aspects of life. Social distancing regulations were enforced in an attempt to halt virus spread. Universities across the country closed for in-person instruction and activities, transitioning to remote learning. University students faced unprecedented challenges and stressors, especially Asian American students due to COVID-19-associated xenophobic attitudes, harassment, and assault against people of Asian complexions. The purpose of this study was to examine the experiences, coping, stress, and adjustment of Asian American students during the COVID-19 pandemic. Secondary analyses were conducted on the survey responses of 207 participants (*n* = 103 Asian American university students, *n* = 104 non-Asian American students) from a larger-scale study, which focused on adaptation to the university, perceived stress, ways of coping, and COVID-19-specific factors. A series of independent samples t tests and regression analyses showed significant relationships between some university adjustment factors, ways of coping methods, and race with perceived stress and COVID-19 factors. Implications, limitations, and ideas for future directions in research are discussed.

## 1. Introduction

On 30 January 2020, the World Health Organization declared a global health emergency, and on 31 January 2020, the United States declared a public health emergency. On 11 March 2020, COVID-19 (“coronavirus disease 2019”) was characterized as a pandemic [1,2]. The COVID-19 pandemic impacted all aspects of life globally via health concerns and the social distancing policies implemented in an attempt to halt virus spread; and university students faced particularly unprecedented stressors [3,4,5,6].

### 1.1. School Closures and Social Distancing

School closures reduce social contact among students, faculty, and staff, and therefore interrupt transmission of disease [7]. In response to the COVID-19 regulations, after the end of the Spring 2020 semester, over 4234 colleges and universities across the United States had been impacted by the effects of COVID-19 [8]. Like the rest of the population, university students experienced the psychological effects of social distancing. However, effects of physical distancing could have been exacerbated in the university student population due to their higher rate of psychological distress [5]. A survey during the early stage of the pandemic of 2086 students found that 80% surveyed reported that the pandemic negatively affected their mental health, with one in five students stating that their mental health significantly worsened [3,5]. Additional challenges were related to lack of independence, being back home for an extended and unexpected period of time, social isolation, and high conflicts and tension [4,5,6].

### 1.2. Xenophobia toward Asian Americans

Xenophobia is defined as dislike of or prejudice against people from other countries [9]. The epicenter of the COVID-19 outbreak was reported to be in Wuhan, Hubei Province, China, in November 2019, and then spread globally [10]. The COVID-19 fueled xenophobic attitudes and actions against people of Asian complexions. Early on during the pandemic, the FBI predicted that “hate crime incidents against Asian Americans likely will surge across the United States, due to the spread of coronavirus disease… endangering Asian American communities” [11]. Asian American university students frequently experience racial prejudice and discrimination, even before the pandemic, and previous research has found that higher rates of depression were reported in Asian American college students who experienced discrimination [12,13,14]. For instance, studies have found that experiences of racial microaggressions predicted increased somatic symptoms and negative affect, such as distress, anxiety, and depression [15], and that racial microaggressions were positively associated with depressive symptoms [16].

Although Asian Americans’ racialized experiences did not begin as a result of the pandemic, COVID-19 fueled increased xenophobic and anti-Asian attitudes and reactions in the public toward Asian children and adults, including verbal and physical abuse, physical hardship, and tremendous emotional burden. Those who were not personally subjected to acts of hate and violence still experienced mental exhaustion from being surrounded by the news of such incidents, in addition to the overall stress and anxiety caused by the COVID-19 pandemic [17].

### 1.3. University Adjustment

Student success in university adjustment correlates with other successes in life [18]. University students display higher rates of depression, anxiety, eating disorders, and other forms of psychological distress than the rest of the population [5,19]. University success is best measured with a variety of components in addition to academic achievement and cognitive capacity, such as academic adjustment, social adjustment, personal–emotional adjustment, and university attachment [20,21,22]. Academic adjustment relates to students’ ability to cope with various educational demands of the university experience, which correlates with students’ academic goals, self-appraisal, and feeling of control over the outcome of their academic efforts [22,23]. Social adjustment relates to students’ ability to cope with interpersonal social demands of the university experience, such as societal activities and functioning, relationships, homesickness, and the university social environment [23]. Personal–emotional adjustment consists of students’ psychological and physical wellbeing and is associated with psychosocial coping skills and resources, degree of experienced psychological distress, and emotional reliance on others [23]. Attachment relates to students’ sense of belonging and their degree of commitment to educational–institutional goals and attachment to their university. Attachment is associated with students’ overall satisfaction with the university experience and drop-out rate [23].

### 1.4. Stress

Stress occurs when it is perceived that the demands of a situation are beyond one’s own capabilities to deal with the circumstances [24]. Perceived stress is how an individual feels about the general stressfulness of their life and their ability to handle such stress during a specific time, such as during the COVID-19 pandemic. Research has found that perceived stress is negatively associated with self-efficacy. Additionally, individuals who perceive a stressful situation as an opportunity to prove themselves use coping skills more effectively and are less likely to think negatively [25,26]. Although university students experienced the same event (i.e., the COVID-19 pandemic), it is each student’s situational perception that dictated their stress.

### 1.5. Coping

According to Lazarus and Folkman’s transactional model of stress and coping [24], personal and situational factors influence how one perceives encountered situations. Variables within a person and the environment affect stress appraisal and coping strategy usage, resulting in physiological and emotional reactions [24]. Coping styles affect the consequences of stressful events as well, such as level of anxiety and psychological distress experienced [27,28].

There are different categories of coping strategies. Emotion-focused coping (i.e., reactive) refers to attempting to regulate feelings and emotional response to the stressor (e.g., anxiety, fear, sadness, anger [29]). Problem-focused (i.e., proactive) coping refers to acting on the stressor, environment, or oneself, addressing the problem, in an attempt to decrease or eliminate the stress [29]. Problem-focused coping is more effective in controllable stressful situations, but emotion-focused coping is more effective in uncontrollable stressful situations [27,30]. Avoidance-focused coping refers to cognitions and behaviors aimed at avoiding the stressful situation and reactions to it (i.e., distraction, diversion), which tends to be an initial reaction to stress [27,28,31,32].

Research has found cross-cultural evidence for the reliability and validity of Lazarus and Folkman’s coping measurement [33,34]. However, Chang [33] examined the coping strategies of Asian American and Caucasian American college students and found that, although two groups were similar in reported coping strategies and behaviors, Asian American college students reported more avoidance strategies and expressed less usage of constructive thinking coping skills. Statistics have also shown that Asian American college students are less likely to seek out support services than students of any other racial group [9,30], which can be attributed to cultural differences and traditional values (e.g., collectivistic versus individualistic [33,35,36]).

### 1.6. Purpose of This Study

While there have been research studies examining adjustment and perceived stress among college students, few studies have examined the similarities and differences between Asian and non-Asian American university students, especially during a public health emergency. This study examined the experiences and ways of adjustment, coping, and perceived stress of university students during the COVID-19 pandemic, specifically those of Asian American undergraduate and graduate students. This study aimed to find averages, patterns, and relationships between various variables related to students’ university adjustment, stress, ways of coping, and COVID-19 adjustment, comparing those of Asian American university students and non-Asian American university students during the early stage of the COVID-19 pandemic in the United States, from April to May 2020. Limited research has examined the difference of university adjustment and coping mechanisms of Asian samples with other-race samples. Although some previous research on university adjustment and coping found similar performance in participants of different races [34,37,38], other research specifically interested in the differences of Asian and comparison samples found that Asian samples differed from comparison samples on aspects of university adjustment, such as personal–emotional adjustment [39] and coping tendency (i.e., tendency to use more avoidance coping strategies [33]).

### 1.7. Theoretical Foundations and Hypotheses

The current study is built upon a number of theoretical foundations. First, in terms of university adjustment, adjustment to the demands of the university is multifaceted (i.e., academic, social, and personal–emotional adjustment and university attachment), according to the theory developed by Baker and Siryk [20,21,22]. University adjustment is associated with success, not only in academic achievement, but in the overall college or university experience. In terms of stress, when one has difficulty dealing with the demands of a situation based on their own capabilities, then one would perceive stress, according to Lazarus and Folkman [24]. In terms of coping approaches, based on Lazarus and Folkman’s transactional model of stress and coping [24], this study was focused on emotion-focused, problem-focused, and avoidance-focused coping strategies.

University adjustment and usage of coping strategies are known to correlate with the ability to adaptively deal with stressors [23,27,28,40]. Grounded in the reports of hate crimes and racism, including the COVID-19-associated xenophobic attitudes and discrimination toward the Asian community, this study hypothesized that: (1) the mean university adjustment and ways of coping scores considering the pandemic would differ between Asian American and non-Asian American university students; (2) Asian American students would report significantly higher perceived stress scores than their counterparts; (3) while considering race (i.e., Asian vs. non-Asian), university adjustment and ways of coping would predict stress in university students during the COVID-19 pandemic, such that those who reported higher university adjustment and those who reported higher coping usage would also report lower perceived stress; and 4) race, university adjustment, and ways of coping would predict adjustment in COVID-19 related factors, with Asian American students reporting lower adjustment related to COVID-19 related factors (i.e., less well-adjusted) due to their unique experiences compared to the non-Asian American sample (i.e., discrimination, [15,41,42]).

## 2. Materials and Methods

### 2.1. Ethical Considerations

In accordance with guidelines for conducting ethical research outlined by the American Psychological Association [43], Institutional Review Board (IRB) approval was obtained from Fordham University for this secondary quantitative study. Data analyses were conducted on a subsample of the original larger-scale survey data that were collected during the early stage of the COVID-19 pandemic in the United States, from April to May 2020. Participants in the current study were recruited for the original larger-scale study through convenience sampling, via professional networks of professors and students. For the original larger-scale study, the goal was to capture university students’ experiences during the initial outbreak of the COVID-19 pandemic; thus, the data were collected from April to May 2020 when the COVID-19 was just declared as a national public health emergency. Participation in the original study was voluntary, and all participants provided consent electronically prior to participating in the study. Study participant IDs (e.g., 001, 002) were assigned to each participant and were associated with all data. No identifiable information was recorded or saved with the data.

### 2.2. Procedures

Potential participants received an invitation to participate in the study with the link to the 25-minute Qualtrics survey that included the study’s consent form and unlabeled self-report measures. Only responses indicating that the survey was completed in its entirety were accepted for the purposes of the original larger-scale study. After completing the survey, participants could elect to enter a raffle to win a USD 10 Amazon gift card (chance of 10:100). For the current study, a subsample of the original data was analyzed specifically to compare the experiences of Asian American and non-Asian American students during the early stage of the COVID-19 pandemic.

### 2.3. Participants

Email invitations were sent to many university student organizations, clubs, and social media groups through the researchers’ professional and personal networks. Professors and administrators in different disciplines and at different universities were contacted to distribute the recruitment information. Eligible participants of the original study were university students who were at least 18 years old and enrolled in an American university at either the undergraduate or graduate/professional level during the initial peak of the COVID-19 pandemic. Power analyses were conducted using the G*Power3 application [44] to determine the appropriate number of participants per group for statistical tests examining mean differences (f = 0.025, power level = 0.80, α = 0.05; *n* ≥ 102) and global effects of multivariance (f = 0.15, power level = 0.80, α = 0.05; *n* ≥ 40). Using the American Council on Education’s reported race make-up of university students in America [45], a representative sample of *N* = 207 participants was included for analyses in the current study (*n* = 103 Asian American university students, *n* = 104 non-Asian American students). No international students were included.

### 2.4. Measures

#### 2.4.1. Demographic Questionnaire

Participants of the original study completed a demographic questionnaire to gather information regarding their race, gender, age, and family income. Relevant university and academic information was also collected.

#### 2.4.2. Student Adaptation to College Questionnaire (SACQ)

The Student Adaptation to College Questionnaire (SACQ; [23]) was used to measure how well students adapted to the demands of the university experience. The SACQ is a 67-item, 9-point Likert scale (“doesn’t apply to me at all” to “applies very closely to me”) questionnaire, resulting in four subscales. The academic adjustment subscale (α = 0.81–0.90; original larger-scale study (OLS) α = 0.88) measured students’ coping with various educational demands of the university experience. The social adjustment subscale (α = 0.83–0.91; OLS α = 0.91) measured students’ coping with interpersonal–societal demands of the university experience. The personal–emotional adjustment subscale (α = 0.77–0.86; OLS α = 0.87) measured students’ psychological and physical wellbeing. The attachment subscale (α = 0.85–0.91; OLS α = 0.90) measured students’ degree of commitment to educational–institutional goals and attachment to their university [20,23]. Participants’ responses for each subscale were totaled, with appropriate items reversed scored, and then converted into T-scores indicating participants’ level of adjustment.

#### 2.4.3. Perceived Stress Scale (PSS)

The 10-item, 5-point Likert scale (“never” to “very often”) Perceived Stress Scale (PSS) was used to measure participants’ perception of stress during the early stage of the COVID-19 pandemic ([46]; α = 0.84–0.86; OLS α = 0.87). The general nature of the questions made them relatively free of specific content for any subpopulation group. The PSS was designed to explore how unpredictable, uncontrollable, and overloaded respondents find their lives to be in the last 30 days. Participants’ scores were summed to calculate a perceived-stress score with higher scores indicating higher level of perceived stress [46].

#### 2.4.4. Ways of Coping Questionnaire (WAYS)

The Ways of Coping Questionnaire (WAYS; [25]) was used to measure participants’ coping processes during the early stage of the COVID-19 pandemic. The WAYS, one of the most widely used coping measures [47,48,49], is a 66-item, 4-point Likert scale (“does not apply or not used” to “used a great deal”) questionnaire (α = 0.78; [50]). Participants’ responses were summed to reveal a score for each of the eight subscales, with higher scores indicating more often usage of behaviors of that subscale in coping with COVID-19. The WAYS categorizes multiple dimensions of coping into emotion-, problem-, or avoidance-focused efforts.

For the purposes of this study, one subscale of each coping type was chosen [28,29]. The positive reappraisal (emotion-focused; α = 0.79; OLS α = 0.74) subscale measured participants’ efforts to create positive meaning by focusing on personal growth. The planful problem solving (problem-focused; α = 0.68; OLS α = 0.71) subscale measured participants’ deliberate problem-focused efforts to alter the situation. The escape avoidance (avoidance-focused; α = 0.72; OLS α = 0.67) subscale measured participants’ wishful thinking and behavioral efforts to escape or avoid the situation [28,51]. Relative scores were calculated to describe the proportion of effort represented by each subscale coping mechanism, with higher relative scores indicating usage of those coping behaviors more often than the other coping mechanisms.

#### 2.4.5. COVID-19 Related Questionnaire

A 37-item, 5-point Likert scale (“strongly disagree” to “strongly agree”) was created for the original larger-scale study to measure the effect of COVID-19 on participants in six subdomains (see Appendix A). This COVID-19 measure was an adaptation of an unpublished instrument created to measure the mental health index and experiences of university students during the initial outbreak in China [52]. Participants’ COVID-19 related responses were scored following the scoring procedures for SACQ. The emotionality subscale (α = 0.71) measured participants’ ability to deal with emotional thoughts and behaviors toward COVID-19 related stimuli and experiences. The adaptive adjustment subscale (α = 0.69) measured participants’ ability to deal with impact, worries, and stress related to their COVID-19 situation. The social support subscale (α = 0.69) measured participants’ level of satisfaction with received support during the COVID-19 pandemic. The academic adjustment subscale (α = 0.85) measured the degree to which participants felt prepared and motivated to complete academic work and their ability to adjust to remote education as a result of the COVID-19 pandemic. The discriminatory impact adjustment subscale (α = 0.78) measured participants’ acknowledgement and impact of racism as related to COVID-19. Finally, the regulation reaction subscale (α = 0.61) measured participants’ agreement with regulations and restrictions imposed due to the COVID-19 pandemic. High scores on COVID-related subdomains indicated more positive adjustment during the COVID-19 pandemic.

## 3. Results

The current study’s participant demographic information can be found in Table 1. Participant ages ranged from 18–57 years (*M* = 22.71), and 68.93% were undergraduate students. About 61% attended universities outside of the metropolitan NYC area, about 38% of participants were originally living off-campus during the Spring 2020 semester, and about 41% of participants indicated a family income of more than USD 100,000.

Reliabilities of scales and statistical assumptions were checked via methods analyzing normal distribution, skewness, and equality of variances. All subscales with less than 10 items reached sufficient Cronbach’s alpha levels of above 0.50 [53]. An independent sample t test was conducted to investigate whether there is a difference in mean SACQ, WAYS, and perceived stress scores of Asian American and non-Asian American students. Tests for normal distribution found skewness in the non-Asian sample’s SACQ social adjustment and personal emotional adjustment, as well as the COVID-19 factors except for adaptive adjustment. Skewness was also found in the Asian American sample’s COVID-19 discriminatory impact adjustment and regulation reaction, as well as WAYS planful problem solving; hence, the results should be interpreted with caution. Levene’s Test showed that both SACQ and WAYS scores demonstrated equal variances. Overall, Asian American students’ SACQ mean scores did not significantly differ from those of non-Asian American students, with the exception of academic adjustment, t(205) = 2.14, *p* = 0.02. Results indicated that non-Asian American students reported significantly higher on escape avoidance coping on WAYS than Asian American students, t(205) = 2.51, *p* = 0.01. Lastly, the mean scores of Asian American students’ perceived stress did not significantly differ from those of non-Asian American students. Please refer to Table 2 for more detailed results.

A multiple linear regression analysis was conducted to explore if race, SACQ factors, and WAYS factors predicted perceived stress. Skewness was not found in either samples’ perceived stress scores; however, as mentioned, skewness was found in some independent variables. A significant regression equation was found (F(8, 198) = 32.65, *p* = 0.000), R^2^ of 0.57). SACQ academic adjustment and personal emotional adjustment were significant negative predictors and WAYS escape avoidance was a significant positive predictor of perceived stress (see Table 3).

A multivariate regression analysis was conducted to explore if race, SACQ factors, and WAYS factors predicted COVID-19 related factors. Box’s test of equality of covariance matrices found that equality of variances was not demonstrated, F(21, 154,537.17) = 4.14, *p* = 0.00. However, Pillai’s trace found acceptable partial Eta squared statistics given the values and data collected for the study. Levene’s test showed that only half the COVID-19 related factor scores demonstrated equal variances; thus, the following results should be considered with caution.

Firstly, results showed no significant relationships between SACQ social adjustment and COVID-19 related factors. Significant positive relationships with small to large effects were found between SACQ academic adjustment and COVID-19 emotionality (*p* = 0.003) and COVID-19 academic adjustment (*p* = 0.000); SACQ personal emotional adjustment and COVID-19 emotionality and adaptive adjustment (*p* = 0.000), COVID-19 academic adjustment (*p* = 0.022), and COVID-19 discriminatory impact adjustment (*p* = 0.004); and SACQ attachment and COVID-19 social support (*p* = 0.000).

Secondly, results showed some significant relationships between WAYS factors and COVID-19 related factors with small to medium effects. Significant positive relationships were found between WAYS positive reappraisal and COVID-19 discriminatory impact adjustment (*p* = 0.034) and COVID-19 regulation reaction (*p* = 0.043); WAYS planful problem solving and COVID-19 regulation reaction (*p* = 0.022); and WAYS escape avoidance and COVID-19 emotionality (*p* = 0.009), COVID-19 adaptive adjustment (*p* = 0.010), COVID-19 discriminatory impact adjustment (*p* = 0.039).

Finally, results showed significant relationships between race and COVID-19 discriminatory impact adjustment (*p* = 0.000) and regulation reaction (*p* = 0.001), with small to medium effects. For more detailed results, please see Table 4 and Table 5.

## 4. Discussion

This study was designed to investigate the experience of Asian American university students and their adjustment, coping, and stress during the COVID-19 outbreak and pandemic. Particularly, the variables of race, adjustment to university, and ways of coping were analyzed to examine perceived stress and COVID-19 experiences.

### 4.1. Differences in University Adjustment and Ways of Coping

The hypothesis that SACQ and WAYS scores would differ between Asian American and non-Asian American university students was partially supported by the results of this study. The Asian American student sample reported lower academic adjustment on the SACQ than the comparison sample. As a group, Asian Americans have been found to perform better academically than other subpopulations, but they also had a higher level of fear of academic failure and spent more time studying and on academic work than their non-Asian peers [36,54]. The lack of significant differences in the other SACQ subscales is consistent with the literature finding similarities in overall college adjustment across racial groups [37,38,39]. Future research can investigate factors of university adjustment of Asian American students beyond the pandemic to better understand the unique impact of race on university adjustment.

The Asian American sample reported significantly lower use of WAYS escape avoidance coping than the non-Asian American sample. This result did not align with Chang’s finding of more usage of avoidance strategies by the Asian sample [33]. However, Chang’s analysis did not report that the college participants were all facing a similar general stressor, such as the COVID-19 pandemic. Coping literature defines avoidance-focused coping as a typical initial reaction to stress [27]. It is possible that the non-Asian sample was newly dealing with the COVID-19 pandemic at the time of the original larger-scale study’s data collection, while the Asian sample had been more aware and facing the news of COVID-19 due to connections or tracking of the pandemic-related news overseas. The COVID-19 pandemic was declared as a public health emergency in some Asian countries as early as January 2020, but it was not declared a public health emergency in the United States until March 2020. Asian American students with connections to families and relatives in Asia may have been more aware of the outbreak and had more time to prepare for its impacts. Future research can examine the relative scores for coping constructs to understand if there is a difference in mean scores for each type of coping type utilized by Asian Americans compared to other groups, as well as the success of avoidance-focused coping with uncontrollable stressors, such as a global pandemic.

### 4.2. Differences in Perceived Stress

The second hypothesis of the current study was that perceived stress scores of Asian American students differed from non-Asian American students during the COVID-19 pandemic. Results showed no significant difference between the two samples. It is perceived stress, how participants felt about their own ability to handle stressors in their lives during COVID-19, that dictated participants’ stress levels [46]. Results suggested that Asian American and non-Asian American students perceived the overall stressor of COVID-19 similarly.

### 4.3. Adjustment and Coping to Predict Perceived Stress

The third hypothesis that race, SACQ factors, and WAYS factors would predict perceived stress was supported by the results of this study in that, overall, the model was found to be significant with some factors significantly correlated with perceived stress. In addition to the negative correlation of previous university academic adjustment and positive correlation of escape avoidance coping discussed, university personal emotional adjustment also negatively correlated with perceived stress during the early stage of the COVID-19 pandemic. The stress of navigating the unknown and uncontrollable situation of the initial peak of the COVID-19 pandemic was inversely related to previous adjustment to personal–emotional demands and stressors associated with university experiences. Experiences related to personal–emotional adjustment with regard to the university experience may have been beneficial in managing perceived stress related to the COVID-19 pandemic. Future research can also consider using a different stress measure that has more questions to reveal more specific stress perception than PSS, which had only 10 questions related to generic stress [55].

### 4.4. Adjustment and Coping to Predict COVID-19 Related Factors

The final hypothesis that race, SACQ factors, and WAYS factors would predict COVID-19 related factors was partially supported by the results of this study. Results indicated that participants who reported higher university academic adjustment also reported higher COVID-19 related emotionality adjustment and academic adjustment. Additionally, participants who reported higher university personal emotional adjustment also reported better adjustment to COVID-19 in terms of emotionality, adaptive, academic, and discriminatory impact adjustment. As discussed above, previous university personal–emotional adjustment negatively correlated with perceived stress during the onset of the COVID-19 pandemic. Participants who reported higher university attachment also reported higher COVID-19 related social support.

### 4.5. Other Highlights

These significant findings aligned with the literature relating academic adjustment and university attachment to social adjustment, perceptions of academic competence, and ability to cope with different demands related to university success [23,56]. Higher reported levels of academic adjustment have been found to be related to self-set academic goals and feelings of control over the outcome of efforts [22,23]. Results of the current study suggested that attachment to one’s university is related to receiving satisfactory social support, as well as to university students’ ability to deal with the impacts and worries related to the COVID-19 situation. Future research can investigate specific support that students received during the COVID-19 pandemic in order to examine the value of support, as well as to explore social support that can be put in place by universities to best support students during situations that greatly impact students. Universities and faculty can promote ways to support students who may be struggling with adjusting to university demands to foster skills and experiences that help university students better navigate challenges in general.

Results of this study also indicated that participants who reported more usage of emotion-focused coping and avoidance-focused coping adjusted better with regard to some COVID-19 related factors. Participants who reported higher positive reappraisal also reported higher discriminatory impact adjustment, indicating that they put forth efforts to create positive meaning by focusing on positive growth. Participants who reported more usage of escape avoidance coping adjusted better with the COVID-19-related factors of emotionality adjustment, adaptive adjustment, academic adjustment, and discriminatory impact adjustment.

Results of this study found planful problem solving (problem-focused coping) to be significantly correlated only with the COVID-19 regulation reaction factor. Main and colleagues [30] concluded through their study of coping and the SARS 2003 epidemic that with regard to uncontrollable, large-scale stressors, any type of coping helps to reduce distress. With regard to most COVID-19 factors, it is likely that participants found that the coping strategy of planful problem solving would not yield much success. Although at the time of the original larger-scale survey, administration effective treatment and vaccines were not yet available for combatting COVID-19, masking and social distancing were practices available and encouraged to the public. Future research can consider the impact of other types of problem-focused coping, such as confrontive coping, which is related to putting forth aggressive efforts to alter the situation or to consider examining problem-focused coping on specific COVID-19 situations, such as facing anti-Asian sentiment. Research related to the COVID-19 pandemic can consider exploring the effect of problem-focused coping after the availability of vaccines and drugs that have been proven to be effective for the prevention and treatment of COVID-19.

Moreover, results of this study found that race correlated with COVID-19 regulation reaction and discriminatory impact adjustment. The regulation of enforcing the usage of face masks in public was controversial [57]. Masks have been reported to slow the spread of COVID-19 by keeping people who are infected from spreading droplets to others when they talk, cough, or sneeze. Masks also protect the wearer, providing protection from COVID-19 or lessening symptom severity. Some people argued that mask mandates violate civil liberties and personal freedom, igniting protests against masks and other regulations put in place to prevent the spread of COVID-19. Asian American students’ significantly better reaction to COVID-19 regulations may be due to the commonality of face mask usage in the East Asian population [58] or their collectivistic tendencies and values [36]. Future research can consider exploring specific motivations behind the level of regulation compliance during the COVID-19 pandemic.

This study found that Asian American participants scored significantly lower than non-Asian American participants in their adjustment to the COVID-19 Asian xenophobic attitudes. This finding suggests that the racism fueled by COVID-19 did not affect non-Asian American students as it did Asian American students. Future research can consider using a discrimination-experience-specific scale to investigate the impact of COVID-19 discrimination on university students, as well as on the Asian population in the United States in general.

## 5. Implications and Recommendations

The Asian American experience of the COVID-19 pandemic was unique compared to that of non-Asian Americans. This study suggested that the COVID-19 pandemic-associated stressors, university adjustments, and anti-Asian sentiment impacted Asian Americans’ overall experiences, coping, stress, and adjustment during the COVID-19 pandemic, in addition to the health concerns related to COVID-19.

Results found that participants who reported higher positive reappraisal coping also reported higher discriminatory impact adjustment, indicating that they put forth efforts to create positive meaning by focusing on positive growth. In Garza Varela and colleague’s research on positive reappraisal as a stress-coping strategy during the COVID-19 pandemic [59], they found that positive reappraisal was associated with greater psychological wellbeing, as well as a lower risk of developing psychopathology. Similarly, Jin and colleagues [60] found that active coping strategies, such as positive reappraisal, were associated with hope later during this pandemic, with meaning in life serving as a mediator. Mental health professionals can consider the benefits of positive reappraisal coping when clients face such an intense and widespread stressor, such as the COVID-19 pandemic and its associated anti-Asian rhetoric and discrimination.

Findings from the current study aligned with research that explored previous disease outbreaks and epidemics, such as SARS and MERS. Previous research found that the likelihood of mental health concerns, such as depression and anxiety, and stress responses, including anger and fear, as a result of disease outbreaks, can persist for months and years [61,62,63]. Public health professionals, journalists, and government officials should raise awareness of the long-lasting impact of epidemics on mental health. Academic advisors and university-based counselors are encouraged to develop university-level, school-level, and program-level support systems to help university students navigate the obstacles associated with large-scale public health emergencies or natural disasters.

## 6. Limitations

Due to the nuance of COVID-19 and its unique impact, specifically on university students and the Asian American population, a measure was adapted to examine COVID-19-related factors for the original larger-scale study. The developed COVID-19-related questionnaire may not have comprehensively investigated the six domains of the measure. Further development of the COVID-19 measure with additional specific questions and further analyses of the scales could reveal additional data and relationships between the current study’s variables. Future research can consider exploring the experience of Asian American university students compared to the experience of specific subgroups of non-Asian American university students, such as Black American students, considering the events leading to, the rallies of, and the aftermath of the Black Lives Matter movement, and other events during the COVID-19 pandemic.

The SACQ instrument measured the multifaceted college adjustment constructs. Although the measure has been proven to demonstrate consistency and validity [23], the lack of attention to diverse populations in its creation and consideration of factors such as culture in college adjustment prevents generalization of the findings. The SACQ was originally validated with primarily White freshmen students in the United States and has not since been empirically studied with other populations such as Asian students. Future research should consider determining suitability and appropriateness of SACQ for diverse populations, as well as for graduate students [23,38]. The WAYS instrument was used to measure coping processes during the COVID-19 pandemic with two coping types: problem-focused and emotion-focused. However, Lazarus and Folkman’s development [24] did not specifically include other research-based coping types (i.e., avoidance-focused). Future research can consider analyzing the coping scores of additional WAYS subscales to examine their relationship with COVID-19 experiences.

The perception of a stressful situation affects the choice and usage of specific coping strategies [24,27]. Asian Americans differ from non-Asians in their way of experiencing emotionality, such as depression. For example, instead of labeling emotionality as depression or anxiety, Asians tend to mention somatic complaints (e.g., changes in appetite, fatigue, restlessness, headaches [36]). Future research can consider coping measures that acknowledge such group differences, including measures that examine experienced emotionality and somatic symptoms during and after the COVID-19 pandemic to explore differences in the perception of experiences.

The current study specifically examined the adjustment and coping of Asian American students. Those who identify as Asian American are a very diverse population. The US Census Bureau [64] lists five race categories (i.e., White, Black/African American, American Indian/Alaska Native, Asian, Native Hawaiian/Other Pacific Islander, and Hispanic/Latino). The original larger-scale study used these options in the demographic question regarding race identification. Placing all Asians under one category can be problematic not only because Asians from different backgrounds have different histories, cultures, and values, but also, specifically important for the current study, they have different physical complexions [36,65]. Due to the reported epicenter of COVID-19 being in China, racist remarks and actions were directed toward not only Chinese Americans, but also other people with similar East-Asian physical complexity. Additionally, the US Census Bureau separates “Asian” and “other Pacific Islander,” although anti-Asian discrimination and hate crime reports are often reported on Asian American Pacific Islander (AAPI) communities overall. Future research should investigate the difference in discriminatory impact and adjustment of people of East-Asian complexity and Southeast-Asian complexity and consider possible gender differences [66].

The original larger-scale study recruited participants through convenience sampling, which allows for the chance of sampling bias [67]. Participants also voluntarily completed the self-report survey in its entirety at their own time and chosen settings. Students who participated may have had different motivations or accessibility than those who did not complete the original survey. Additionally, although sample participants were pulled for this current study by strategically using official data on the make-up of university students in America [45], it is likely that the participants’ experiences differed based on where they were located within the United States, as well as their other demographic information. Discriminatory impact, as well as overall experiences during the COVID-19 pandemic, could vary based on location due to the differences in Asian American populations by state throughout the United States [68]. Lastly, although self-administrated methods lead to more frequent and accurate reporting of sensitive information [67], it is possible that the accuracy of participants’ self-report might have affected the collected data.

## 7. Conclusions

This study explored and examined the experiences and ways of adjustment, coping, and perceived stress of university students during the COVID-19 pandemic, specifically those of Asian American undergraduate and graduate students. Results suggested that in addition to health-related concerns and effects of social-distancing of the COVID-19 pandemic, Asian American university students’ overall experiences, stress, coping, and adjustment during the COVID-19 pandemic were influenced by the COVID-19 pandemic-related stressors, their prior university adjustment, and the anti-Asian sentiment. This study contributes to the literature, as it sheds light on the unique experiences of Asian American students during the COVID-19 pandemic, in comparison to their non-Asian American counterparts. Future research should expand on this work by exploring Asian American students’ unique experiences in depth and by informing institutions on the necessary supports that should be in place to support students during unprecedented situations, such as the COVID-19 pandemic.

## Figures and Tables

**Table 1 ijerph-20-04162-t001:** Participant demographics (*N* = 207; *n* = 103 Asian American; *n* = 104 non-Asian American).

	Asian		Non-Asian	
	*n*	%	*n*	%
Race				
Asian	103	100	/	/
White	/	/	68	32.85
Black or African American	/	/	8	3.86
Hispanic or Latino	/	/	20	9.66
American Indian or Alaska Native	/	/	1	0.48
Native Hawaiian or Other Pacific Islander	/	/	1	0.48
Other (e.g., Biracial)	/	/	6	2.90
Gender				
Male	18	17.48	31	29.81
Female	85	82.52	73	70.19
Age (years)				
18–25	88	85.44	89	85.57
26–57	15	15.53	15	15.38
Household Income				
Less than USD 20,000	7	6.80	6	5.77
USD 20,000 to 99,999	59	57.28	51	49.04
More than USD 100,000	37	35.92	47	45.19
University Level				
Undergraduate	67	65.05	77	74.04
Graduate/Professional	36	39.95	27	25.96
University Location				
Metropolitan NYC Area	32	31.07	48	46.15
Outside of Metropolitan NYC Area	71	68.93	56	53.85

**Table 2 ijerph-20-04162-t002:** Independent samples *t* test on factors of SACQ, WAYS, and perceived Stress.

		Non-Asian American		Asian American		Levene’s Test F	Sig.	t	*p*	Effect Size
		M	SD	M	SD					
SACQ	Academic Adjustment	48.48	9.33	45.94	9.34	0.05	0.82	2.14	0.02 *	0.30
	Social Adjustment	47.89	9.83	46.02	8.68	0.92	0.34	1.45	1.47	0.20
	Personal Emotional Adjustment	39.28	9.20	40.81	9.30	0.00	0.95	−1.19	0.24	−0.17
	Attachment	50.37	9.71	48.72	7.86	3.16	0.08	1.34	0.18	0.19
WAYS	Emotion-Focused Positive Reappraisal	6.88	3.19	6.45	3.39	0.01	0.92	0.94	0.18	0.13
	Problem-Focused Planful Problem Solving	7.43	3.58	7.17	4.04	0.65	0.42	0.49	0.63	0.07
	Avoidance-Focused Escape Avoidance	11.17	3.99	9.78	4.01	0.02	0.90	2.51	0.01 **	0.35
Stress		21.37	6.52	20.20	5.85	1.61	0.21	1.35	0.18	0.19

* *p* ≤ 0.05, ** *p* ≤ 0.01.

**Table 3 ijerph-20-04162-t003:** Multiple regression of race, SACQ, and WAYS on perceived stress.

		B	SE	95% CI	β	t	Sig.	R^2^	ΔR
Model								0.75	0.57
Race		−0.55	0.61	[−1.74, 0.65]	−0.04	−0.90	0.37		
SACQ	Academic Adjustment	−0.16	0.05	[−0.26, −0.07]	−0.23	−3.39	0.00 ***		
	Social Adjustment	−0.02	0.06	[−0.14, 0.12]	−0.02	−0.25	0.81		
	Personal Emotional Adjustment	−0.33	0.04	[−0.41, −0.25]	−0.49	−8.45	0.00 ***		
	Attachment	0.08	0.07	[−0.05, 0.22]	0.12	1.20	0.23		
WAYS	Positive Reappraisal	−0.09	0.09	[−0.26, 0.08]	−0.06	−1.00	−0.32		
	Planful Problem Solving	−0.13	0.11	[−0.40, 0.03]	−0.10	−1.67	0.10		
	Escape Avoidance	0.39	0.08	[0.24, 0.54]	0.25	4.93	0.00 ***		

*** *p* ≤ 0.001.

**Table 4 ijerph-20-04162-t004:** Multivariate regression of SACQ, WAYS, and race on COVID-19 factors.

Source	Dependent Variable	df	Mean Square	F	Sig.	Partial Eta Squared
SACQ						
Academic Adjustment	Emotionality	1	78.22	9.00	0.00 **	0.04
	Adaptive Adjustment	1	0.06	0.00	0.95	0.00
	Social Support	1	17.48	2.55	0.11	0.01
	Academic Adjustment	1	907.40	39.24	0.00 ***	0.17
	Discriminatory Impact Adjustment	1	9.34	1.86	0.17	0.01
	Regulation Reaction	1	30.47	3.26	0.07	0.02
Social Adjustment	Emotionality	1	11.89	1.37	0.24	0.01
	Adaptive Adjustment	1	4.14	0.24	0.63	0.00
	Social Support	1	25.03	3.66	0.06	0.02
	Academic Adjustment	1	5.57	0.24	0.62	0.00
	Discriminatory Impact Adjustment	1	12.07	2.40	0.12	0.01
	Regulation Reaction	1	3.09	0.33	0.57	0.00
Personal Emotional Adjustment	Emotionality	1	665.12	76.53	0.00 ***	0.28
	Adaptive Adjustment	1	384.59	22.04	0.00 ***	0.10
	Social Support	1	0.06	0.01	0.93	0.00
	Academic Adjustment	1	123.43	5.34	0.02 *	0.03
	Discriminatory Impact Adjustment	1	42.58	8.48	0.00 **	0.04
	Regulation Reaction	1	19.46	2.08	0.15	0.01
Attachment	Emotionality	1	1.17	0.14	0.71	0.00
	Adaptive Adjustment	1	26.71	1.53	0.22	0.00
	Social Support	1	87.42	12.77	0.00 ***	0.06
	Academic Adjustment	1	72.10	3.12	0.08	0.02
	Discriminatory Impact Adjustment	1	1.66	0.33	0.57	0.00
	Regulation Reaction	1	2.53	0.27	0.60	0.00
WAYS						
Positive Reappraisal	Emotionality	1	5.32	0.61	0.44	0.00
	Adaptive Adjustment	1	0.13	0.01	0.93	0.00
	Social Support	1	1.04	0.15	0.70	0.00
	Academic Adjustment	1	0.17	0.01	0.93	0.00
	Discriminatory Impact Adjustment	1	22.81	4.54	0.03 *	0.02
	Regulation Reaction	1	38.60	4.13	0.04 *	0.02
Planful Problem Solving	Emotionality	1	0.03	0.00	0.95	0.00
	Adaptive Adjustment	1	35.76	2.05	0.15	0.01
	Social Support	1	7.20	1.05	0.31	0.01
	Academic Adjustment	1	34.64	1.50	0.22	0.01
	Discriminatory Impact Adjustment	1	9.78	1.95	0.16	0.01
	Regulation Reaction	1	49.92	5.34	0.02 *	0.03
Escape Avoidance	Emotionality	1	60.25	6.93	0.01 **	0.03
	Adaptive Adjustment	1	118.82	6.81	0.01 **	0.03
	Social Support	1	5.31	0.78	0.38	0.00
	Academic Adjustment	1	69.27	3.00	0.09	0.02
	Discriminatory Impact Adjustment	1	21.62	4.31	0.04 *	0.02
	Regulation Reaction	1	0.21	0.02	0.88	0.00
Race	Emotionality	1	11.90	1.37	0.24	0.01
	Adaptive Adjustment	1	57.76	3.31	0.07	0.02
	Social Support	1	13.29	1.94	0.17	0.01
	Academic Adjustment	1	34.89	1.51	0.22	0.01
	Discriminatory Impact Adjustment	1	106.42	21.20	0.00 ***	0.10
	Regulation Reaction	1	99.92	10.70	0.00 **	0.05

* *p* ≤ 0.05, ** *p* ≤ 0.01, *** *p* ≤ 0.001.

**Table 5 ijerph-20-04162-t005:** Pairwise comparisons of race on COVID-19 factors.

	Mean Difference	SE	Sig.	95% Confidence Interval
Emotionality	0.50	0.43	0.24	[−0.43, 1.35]
Adaptive Adjustment	1.11	0.61	0.07	[−0.09, 2.30]
Social Support	−0.53	0.38	0.17	[−1.28, 0.22]
Academic Adjustment	−0.86	0.70	0.22	[−2.24, 0.52]
Discriminatory Impact Adjustment	1.50	0.33	0.00 ***	[0.86, 2.14]
Regulation Reaction	−1.45	0.45	0.00 ***	[−2.33, −0.58]

*** *p* ≤ 0.001.

## Data Availability

Due to privacy concerns mentioned in the IRB protocol, the data associated with this study cannot be provided to the public without the supervision of the researchers. However, individual researchers who are interested in obtaining access to the data for individual use are encouraged to contact the corresponding author.

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
