# Peer review of "Asian American University Students’ Adjustment, Coping, and Stress during the COVID-19 Pandemic"

_ijerph, 2023, doi:10.3390/ijerph20054162_

Round 1

Reviewer 1 Report

1. The title of the article is very broad, but the results of the analysis are very limited. Therefore, I suggested that the title of the paper could be redefined.

2. Please clarify why you are using data from three years ago

3. In the introduction, please describe the gaps in the study.

4. Please add a section on theoretical foundations and research hypotheses. 3.

5. "Participants" should be moved to after "Procedures". 4.

6. The discussion is too general and should be written under subheadings according to the research hypothesis.

7. In the Appendix, please add the full questionnaire.

8. Please provide recommendations based on the post-epidemic era.

9. Please indicate if all participants felt discriminated against.

10. Please add references for 2022 and 2023.

Author Response

Please find the attached file (responses to the reviewer 1). 

Reviewer 2 Report

Dear Authors,

it was a pleasure to review your manuscript. Your research topic is socially significant. The research report is very clear to read and understand, it is fluent and thorough. The background and the research questions are connected in a logical way. The variables chosen and the analyses are in accordance with the vantage point. The result tables are introduced and interpreted properly and also the discussion and conclusions section meet the level of a proper research article.

However, I see that sections 1. Introduction and 2. Materials and methods, need some revision.

1.2.The concept of "xenophobia" should be defined, not taken for granted and the arguments of COVID-19 as fueling xenophobia should be clearly expressed. The current version only states that this type of fueling did happen. The reader cannot know this for sure. One possible reason - the initial COVID-19 outbreak in China (Wuhan) - is mentioned only later in the text with a reference number 48. However, the authors might have other, perhaps even scientific,  evidence for their claim.

2.2. Procedures: The selection process of the potential participants to be included in the convenience sample should be explained in more detail. The authors understand that this sampling methods is vulnerable to bias (reference number 64), but they do not try to convince the reader about the soundness of their actual choices to recruit to participants.

All the best for your revision work and future research!

Sincerely

Your reviewer

Author Response

Please find the attached file (responses to the reviewer 2). 

Round 2

Reviewer 1 Report

Please add references in the third sentence. “Xenophobia is defined as dislike of or prejudice against people from other countries. The epicenter of the COVID-19 outbreak was reported to be in Wuhan, Hubei Province, China in November 2019, and then spread globally. The COVID-19 fueled xenophobia attitudes and actions against people of Asian complexions.

Author Response

Please find the attached letter. 
